# Replicative Aging in Pathogenic Fungi

**DOI:** 10.3390/jof7010006

**Published:** 2020-12-25

**Authors:** Somanon Bhattacharya, Tejas Bouklas, Bettina C. Fries

**Affiliations:** 1Department of Medicine, Stony Brook University, Stony Brook, NY 11794, USA; bouklast@oldwestbury.edu (T.B.); Bettina.Fries@stonybrookmedicine.edu (B.C.F.); 2Department of Biological Sciences, State University of New York College at Old Westbury, Old Westbury, NY 11568, USA; 3Department of Microbiology and Immunology, Stony Brook University, Stony Brook, NY 11794, USA; 4Veterans Administration Medical Center, Northport, NY 11768, USA

**Keywords:** aging, fungal pathogen, replicative aging, drug resistance, persister cells, microfluidics, biotinylation, cell isolation, phenotypic variations, high-throughput techniques, fungal pathogen, yeast

## Abstract

*Candida albicans*, *Candida auris*, *Candida glabrata*, and *Cryptococcus neoformans* are pathogenic yeasts which can cause systemic infections in immune-compromised as well as immune-competent individuals. These yeasts undergo replicative aging analogous to a process first described in the nonpathogenic yeast *Saccharomyces cerevisiae*. The hallmark of replicative aging is the asymmetric cell division of mother yeast cells that leads to the production of a phenotypically distinct daughter cell. Several techniques to study aging that have been pioneered in *S. cerevisiae* have been adapted to study aging in other pathogenic yeasts. The studies indicate that aging is relevant for virulence in pathogenic fungi. As the mother yeast cell progressively ages, every ensuing asymmetric cell division leads to striking phenotypic changes, which results in increased antifungal and antiphagocytic resistance. This review summarizes the various techniques that are used to study replicative aging in pathogenic fungi along with their limitations. Additionally, the review summarizes some key phenotypic variations that have been identified and are associated with changes in virulence or resistance and thus promote persistence of older cells.

## 1. Introduction

Most fungal pathogens cause subacute or chronic infections. One major challenge in designing effective antifungal drug therapy is a knowledge gap regarding the impact of microbial microevolution in the host during chronic infection on antifungal resistance of the respective fungal population. *Cryptococcus neoformans*, *Candida albicans*, *Candida auris*, and *Candida glabrata* are pathogenic yeasts, expand clonally in the host, and similar to the unicellular yeast *Saccharomyces cerevisiae*, they undergo asymmetric mitotic divisions. Throughout this clonal expansion, budding mother cells progressively age after each division, which is a process referred to as “replicative aging (RA)” [1,2]. A mother cell can only produce a finite number of buds during mitotic division, and the total number of buds that a mother cell produces before it stops dividing and dies is the designated replicative life span (RLS) of that yeast cell. Each time a mother cell produces a bud, it becomes one generation older (Figure 1a). During RA, mother cells undergo unique phenotypic changes that set them apart from their respective daughter cells [3,4,5]. In an exponentially growing pathogen population, the older mother cells will become increasingly scarce with cumulative divisions that produce young daughter cells, as described in Figure 1b. For example, the frequency of a 10-generation-old (GEN 10) mother cell in a growing population is only 1 in 1024 cells, whereas the frequency of a GEN 2 mother cell is 1 in 4 cells. There is now a set of published data that indicates that older mother cells of several pathogenic fungi, including *Candida glabrata*, *Candida auris*, *Candida albicans*, and *Cryptococcus neoformans*, exhibit enhanced resilience to antifungals and oxidative stress, which could potentially give them an advantage in the host environment. Besides enhanced antifungal resilience, generationally aged cells from pathogenic yeasts are tolerant to phagocytic killings by macrophages. However, these phenotypic variations during RA in pathogenic yeasts are not observed in generationally aging *S. cerevisiae* cells [6]. Thus, it is important to study RA in pathogenic yeasts. This review summarizes the various techniques that are used in studying RA and also summarizes critical phenotypic variations that are associated with aging in pathogenic fungi.

## 2. Methods to Analyze Replicative Life Span (RLS)

Methods to investigate aging were first pioneered in *S. cerevisiae* beginning 1959 [7]. Most of these techniques are over 20 years old [7]. Some of these methods have now been adapted to study replicative aging in pathogenic fungi. Five methods are described for analyzing RLS in yeast. These include (i) the microdissection method (ii) the high-throughput microfluidics method, (iii) biotin label and fluorescence-activated cell sorting (FACS)/magnetic sorting, (iv) centrifugation, and (v) the mother enrichment program (genetic mutation). Table 1 summarizes the techniques that are used to isolate cells of advanced generational age in several pathogenic and nonpathogenic yeasts.

### 2.1. Microdissection Method

Microdissection is the gold standard method to analyze RLS. This method requires the use of a dissection microscope, which includes a microneedle. The microneedle is a 50 µm fiber optic needle with the help of which around 30–40 naïve yeast cells are lined up on a culture plate (Figure 2a) [8,9]. The microneedle then helps to separate buds from these naïve cells after the cells replicate every 1–2 h. These budding events are observed through the microscope, manually scored, and the total number of buds that a mother cell produces before senescing is recorded and used to calculate the RLS of the cell. This method requires manually removing buds from their respective mother cells after every budding event. So, each day, 6–8 budding events are recorded. At the end of the day, the culture plates containing the naïve cells are stored at 4 °C to slow down cell replication, and the next day buds are again separated to continue with the RLS analysis. Using this method, RLS analysis is completed in one to four weeks.

Limitations using this method to determine RLS are as follows:The method is labor intensive and time consuming. As mentioned above, RLS analysis can take as long as four weeks. Further, this method creates a discontinuity in the analysis, as at the end of the day the culture plates are refrigerated to slow down the replication time. This can cause unnecessary stress on the cells resulting in erroneous RLS determination.Since this method is time consuming and often lasts one to four weeks, the culture media can degrade or get contaminated, causing defects in cell growth affecting RLS.Only a small number of cells (around 30–40) can be used to analyze RLS. RLS varies from cell to cell within the same strain, which is referred to as stochasticity of life span. Hence, since a large number of cells are required to accurately analyze the stochasticity of RLS, this method is not ideal for that analysis.This method requires proper identification of mother and daughter cells after each budding event. For the first few generations, both mother and daughter cells are similar in size. This makes identifying daughter cells difficult, causing erroneous RLS analysis.

### 2.2. High-Throughput Microfluidics Method

To overcome the outlined disadvantages of the microdissection method, microfluidic systems were designed by several laboratories [8,10,11,12,13,14]. They represent the most recent innovation in the field of replicative aging research and permit high-throughput investigations. This technique allows the study of single-cell RA [15]. The microfluidic system contains a single channel with inlets and outlets (Figure 2b). The continuous flow of growth media is maintained to ensure that cells remain trapped and media flows through the inlets with the help of syringes and syringe pumps as described previously [8]. The system consists of a microscope slide in which 10 individual channels are aligned. Each channel consists of 80 rows of microbuckets. These microbuckets individually trap a single cell. The buckets permit the trapped cells to bud in a variety of ways, and these buds are washed away by the continuous flow of media. Within these buckets, the trapped single cell grows in size as it progressively ages. The buckets are designed to retain these growing cells. However, some cells can outgrow the buckets and can get swept away. For a microfluidics system designed for *C. neoformans*, it was reported that 62% of the cells could be retained in the buckets throughout their entire life span [8].

The microfluidic technique is high throughput, and RLS can be determined in hundreds of cells simultaneously. Since so many cells can be analyzed for RLS at the same time by this method, the method permits modeling of RLS stochasticity and variance within a population. This technique is semiautomated, as each budding event is recorded by a video, and the number of budding events can retrospectively be determined and RLS can be calculated. The process of determining the actual RLS is continuous with this method, thereby shortening the experimental time significantly. This method can potentially also improve the accuracy of RLS, as the mother cells are not subjected to unnecessary stress from constant temperature changes, to which mother cells are exposed with the conventional method.

The microfluidic technique has the following limitations:Extreme longevity RLS is difficult to assess with the help of this device. The device may get clogged due to a cell overgrowing during prolonged runs. Further, the method requires a continuous supply of fresh media through syringes. The volumes of syringes are limited, and if the cells have excessively long life spans, the device may run out of fresh media. As a result, buckets may get clogged.The microscope is sensitive to any liquid spills, so the system needs to be continuously monitored to prevent any leaks.Any extreme phenotypic change (hyphal formation or titan cell) will clog the system and abort the recording of RLS on that specific cell.There are still no reliable programs that permit automatic determination of bud count and RLS computation. Budding events are therefore still counted manually by reviewing the video images.

### 2.3. Biotin Label and Magnetic/FACS Sorting

Each time a yeast cell divides, it produces a bud that eventually separates from its mother and becomes a daughter cell. The cell surfaces of the daughter cells are newly synthesized and do not share any cell surface components with the mother cells. Therefore, the mother cell surface proteins can be tagged, and the tag can be used to isolate the mother cells from a mixed culture containing mother and daughter cells. In this isolation method, cell surface proteins are first conjugated with biotin, and then biotinylated cells are grown in liquid cultures for the desired number of generations. After cell growth, fluorochrome-conjugated streptavidin is added to the culture. The fluorochrome attaches to the biotin-labeled old cells and can be separated using fluorescence-activated cell sorting (FACS). This method allows rapid purification of the old cell population; however, the total yield of the old cells is low (good for only isolating 10^4^ cells).

Due to low yield in FACS, the method was further modified to improve yield by using streptavidin-coated paramagnetic iron beads instead of the fluorochrome. The beads coat the old cells which are eventually separated by passing them through a column attached to a magnet. The old labeled cells adhere to the column, while young unlabeled cells pass through. The older cells are retrieved once the magnet is removed. This method is very useful since it combines the purity of biotin-dependent sorting with the ability to sort a larger number of cells. This method allows for the isolation of 10^8^ cells. Magnetic sorting has been extensively used in *S. cerevisiae*, *C. glabrata*, *C. auris*, and *C. neoformans* [16,17,18,19]. The technique is described in Figure 3a.

Limitations of these large-scale isolation methods are as follows:Encapsulated fungi such as *C. neoformans* may shield the biotin-labeled cell surface proteins, which ultimately impairs the binding of streptavidin-conjugated beads, leading to low yield of older mother cells.Expense is considerable, especially when yield remains low.In *S. cerevisiae,* the age of the isolated population can be easily verified. *S. cerevisiae* produces permanent scars in the cell wall each time they produce daughter cells. These scars can be visualized by staining with calcofluor white, and the generational age of the isolated population can be determined by counting the bud scars. However, cell wall scars heal in *C. neoformans* and any nonhealed scars are obscured by capsules. In *C. glabrata*, the bud scar count is more reliable.

### 2.4. Centrifugation

#### 2.4.1. Sucrose Gradient Centrifugation

Each time a mother cell produces a daughter cell, it increases in size. After a few generations, the mother cells become substantially bigger than their respective daughter cells. This size difference can be exploited by sucrose gradient centrifugation to isolate cells of advanced age. This technique uses 10–30% sucrose gradient to separate older (bigger) mother cells from younger (smaller) daughter cells. The sucrose gradient separates the mixed population into two distinct bands of cells, one of which is mostly comprised of young cells. These young cells are then exposed to mating pheromones and allowed to replicate for at least three generations. Daughter cells are separated from the mother cells by repeating the sucrose gradient, and this time the band of cells containing the older cell population is selected. This cycle is repeated to isolate cells of a desired generational age with as little as 10% impurity. Thus, this technique allows relatively pure large-scale isolation of older generation cells. This technique has been used for the isolation of generationally aged *C. albicans* cells [20].

Limitations of this technique are as follows [19]:The technique is labor intensive and requires multiple rounds of manipulation to isolate cells of the desired age.

Since the above technique is labor intensive, another centrifugation method of studying RA was designed that is described below.

#### 2.4.2. Centrifugal Elutriation

This is another technique that relies on the cell size difference between mother and daughter cells for isolation of old cells. The specialized elutriation rotor contains chambers in which cells are grown and eluted based on size and sedimentation during centrifugation. This technique can be used to continuously separate daughter cells from mother cells. Isolated daughter cells are collected in different fractions and subjected to a second round of elutriation during which the daughter cells are discarded and mother cells are retrieved. This process is continuously carried out for several divisions to isolate cells of the desired age. This technique allows the rapid isolation of older cells and is less labor intensive than sucrose gradient centrifugation, requiring fewer manipulations. This technique is cheap if the specialized centrifuge is available. It has been commonly used in *S. cerevisiae*, *C. neoformans*, and *C. glabrata* [18,21,22].

Limitations of this technique are as follows [19]:Elutriation does not often deliver a pure population if used repeatedly (cells older than 10 generations).Further, elutriation is not reliable for fungal populations that exhibit cell size heterogeneity at baseline.

The limitations of both types of centrifugation techniques still exist. Hence, presently, the other techniques are preferred in studying RA in pathogenic yeasts.

### 2.5. Mother Enrichment Program (Genetic Mutation)

This highly innovative technique uses an inducible genetic system in which mother cells maintain a normal replicative life span, while the daughter cells are eliminated [23]. This technique was first developed and applied to study age-related phenotypes of *S. cerevisiae* [24,25,26] and does not require any micromanipulations. The mother enrichment program (MEP) uses a Cre-lox recombinase system to knockout two essential genes—*UBC9* (encoding SUMO-conjugating enzyme) and *CDC20* (encoding activator of the anaphase-promoting complex)—in the daughter cells. Both genes are required for cell cycle progression and disrupting them results in permanent M-phase cell cycle arrest in the daughter cells. The Cre recombinase system is regulated by a daughter-specific promoter *P_SCW11_*, which is expressed by transcription factor Ace2p. Ace2p is a transcription factor that is asymmetrically distributed in the daughter cell nuclei before cytokinesis. In this system, an estradiol-binding domain (EBD) is also fused, which post-transcriptionally regulates the Cre activity in the presence or absence of estradiol. In the presence of estradiol, the fusion protein is transported to the nucleus where the protein acts on *loxP* DNA substrates [23]. This results in the elimination of young cells from a growing culture while the mother cells continue to replicate. The system allows cheap isolation of large numbers of old cells and has been used in recent years to gain insights into important aspects of replicative aging, such as its causal relationship with genomic instability [27]. This technique is schematically described in Figure 3b.

Limitations of MEP are as follows [23]:The technique requires genetic manipulations that need to be introduced in the desired strains. This can be difficult in organisms such as *C. neoformans*, which are more difficult to transform than *S. cerevisiae*.Mutations in MEP strains can arise that can prevent the selection of the transformants.The technique is labor intensive.Purity has to be verified. In *S. cerevisiae*, it was found that the M-phase-arrested daughter cells do not senesce immediately, were metabolically active, and continued growth in the arrested state for at least 24 h. These cells produced 6.9 progenies before lysing in the presence of estradiol.

## 3. Phenotypic Variations Associated with Aging

### 3.1. Phenotypic Differences between Mother and Daughter Cells in Unicellular Yeasts

Age-related phenotypic changes can differ among *Saccharomyces cerevisiae*, *Cryptococcus neoformans*, *Candida albicans*, *Candida glabrata*, and *Candida auris*, including those involving cellular morphology (cell size, cell wall thickness, cell shape during division, and arrest) and cytosolic components (extrachromosomal rDNA circles, free radicals, and melanin) (summarized in Table 2). Phenotypic differences of yeast mother cells and their respective daughters have been well described in *S. cerevisiae* [30]. Cell size increases in these yeast cells during aging [31,32,33]. The small surface-area-to-volume ratio in older cells confers resistance to drugs. This evolutionary advantage is not limited to eukaryotes, as even bacteria with increased volume exhibit protection against toxic solvents or antibiotics [34,35]. For *S. cerevisiae*, resilience against some stresses appears early during replicative aging. For example, mother cells with 1–3 bud scars are more resistant to acetic acid but not to other types of stress, such as heat shock [32,36], compared to daughter cells. It is noteworthy that data derived from the *SIR2* mutant collection in *S. cerevisiae* indicate that small cell size at the beginning of a life span is associated with a long life span [37]. This correlation was not confirmed in *C. neoformans* strains with different cell sizes, where cell size did not predict RLS extension [38]. In *C. neoformans*, sirtuin mutants have a strain- and calorie-dependent effect on life span that was observed by chemical modulation rather than genetic deletion [39,40].

*C. neoformans* is encapsulated, which independently affects the cell size [21,41]. Moreover, *C. neoformans*’ polysaccharide capsule is further induced under specific environmental conditions, such as low CO_2_, low iron, and higher temperatures. Hence, in this yeast, cell size variability is contributed from cell body size change as well as capsule. It is noteworthy that variable cell body sizes have been documented in murine [42], rat, and human cryptococcal infections [6] and gives further support to the notion that older cells accumulate in vivo.

Marked phenotypic variation in cellular morphology has been observed in *Candida* species of advanced generational age. During aging, *C. albicans* produces long hyphae [20] and *C. glabrata* produces pseudohyphae [18], whereas *C. auris* makes no pseudohyphal protrusions and remains relatively small even with advanced generational age [17,43]. For *Candida* species, the ability to grow into pseudohyphal and hyphal forms under inducing conditions, which include high temperature, serum, CO_2_ and O_2_ tension, and neutral pH, has been implicated in better invasion of epithelial cells and increased tissue damage [44]. Whether replicative aging-induced morphological transitions contribute to enhanced virulence remains to be resolved.

The fungal cell wall is a complex and flexible structure composed of chitin, α- and β- linked glucans, glycoproteins, and in some cases pigments such as melanin [45]. However, the specific cell composition and structure differ among the fungi. Fungal cell walls protect the cell from different types of stress, including osmotic stress, and exhibit considerable plasticity also throughout aging. In *S. cerevisiae*, the cell wall weakens as increasing numbers of bud scars accumulate throughout replicative aging [19,46]. In contrast, cell wall thickness is a hallmark of older pathogenic yeasts. The association of cell wall change and augmented antifungal resistance was observed in older *C. neoformans* [6,21], *C. glabrata* [16,18], and most recently *C. auris* [17,43] cells. In older *C. neoformans* cells, remodeling of the cell wall with more abundant beta-glucans has been described [47,48]. In 10-generation-old *C. neoformans* cells, accumulation of antiphagocytic protein (App1), which is a key virulence factor [29] located in the cell wall, is observed.

In *C. auris*, generationally older cells also exhibit a thickened cell wall and enhanced adhesion in cell culture, which is also supported by upregulation of several conserved adhesin proteins [17]. In *C. albicans* and *C. glabrata*, epithelial adhesion proteins that inhabit the cell wall may be differentially exposed from cell wall thickening [49]. This is important for *Candida* species, where increased phagocytosis and inflammatory response are associated with increased adhesion [50].

Plasma membrane proteins [51,52,53] and protein aggregates that sustain oxidative damage [54,55] have been shown to be inherited asymmetrically during replicative aging in *S. cerevisiae*. In *C. neoformans*, *ALL2* helps to maintain intracellular pH and is upregulated under low glucose conditions similar to what the pathogen encounters in the host environment. Interestingly, *ALL2* also accumulates during aging under low glucose conditions [8]. In *S. cerevisiae*, glycogen accumulation and upregulation of genes involved in glycogen production have been associated with aging mother cells [56,57] and have been hypothesized to occur from a turnover of carbohydrate metabolism. This is consistent with data from *C. albicans*, where increasing levels of glycogen have been observed with increasing age [20]. Further, in *C. albicans*, a high number of carbonylated proteins have been shown to accumulate with age [20], and this is consistent with data reported from older *S. cerevisiae* cells [54,58].

Melanins are biologically prominent macromolecules that are formed by oxidative polymerization of phenolic compounds. They have been linked with virulence in several pathogenic fungi, especially *Cryptococcus neoformans* [59,60,61,62,63]. There, melanins are synthesized within the cell wall from quinones and contribute to the resistance against phagocytosis, free-radical killing, and even antifungal resistance to the antifungals amphotericin B and caspofungin [64,65,66]. In *C. neoformans,* a disproportionate inheritance of melanin in mother cells has been established [67]. Notably, melanin variants enriched in select lipids (sterol esters and poly-isoprenoids) were recently suggested to be linked to cell aging [68]. The disproportionate inheritance is supported by upregulated levels of *LAC1* and *LAC2*, which contribute to melanin formation, in 10-generation-old *C. neoformans* cells [29]. *LAC1* is widely conserved in *Candida* and its role in age-related phenotypic variation should be examined further in *Candida* species, especially in the emergent *C. auris*.

### 3.2. Antifungal Resistance in Older Fungal Cells

In *S. cerevisiae*, age-related changes of MDR protein expression between old mother and young daughter cells are reported. Similarly, 10-generation-old mother cells from *C. neoformans, C. glabrata,* and *C. auris* also show altered MDR expressions that corelate with increased tolerance to the antifungals fluconazole and amphotericin B [6,16,17,18]. Additionally, increased micafungin tolerance was also observed in 10-generation-old cells from *C. glabrata* and *C. auris* [16,17]

To date, the mechanisms of increased tolerance to fluconazole in old mother cells have only been studied in *C. glabrata* and *C. auris.* In these yeasts, fluconazole tolerance corelated with increased expression of genes encoding the fluconazole drug target 14α demethylase (Erg11p) in the older mother cells. Besides the drug target, overexpression of membrane transporters was also observed in the mother cells from *C. glabrata* and *C. auris* [16,17]. Both membrane transporters adenosine triphosphate (ATP) binding cassette transporters (ABC-T) and major facilitator transporters (MFS-T) are attributed to fluconazole resistance in several pathogenic fungi [69]. Several ABC-Ts, specifically Cdr1p, Ybt1p, and Pdr16p, were upregulated in *C. glabrata* 14-generation-old cells [16] when compared with the younger daughter cells. ABC-T, Cdr1p was also upregulated in aging mother cells of *C. auris* [17]. Interestingly, in *C. auris*, it was found that the observed increased expression of *CDR1* was associated with an increased copy number of *CDR1* in the older mother cells [17]. It is not clear yet if the observed increased copy number of *CDR1* resulted from whole chromosome duplication. Importantly, it was observed to be transient and not observed in the daughter cells that budded off the mother cell. Further investigation with whole-genome sequencing needs to be undertaken once the *C. auris* genome is more completely curated.

### 3.3. In Vitro and In Vivo Virulence

Of particular interest is the change in virulence for pathogenic fungi during aging. Virulence changes do not apply to *S. cerevisiae*, as this yeast is only rarely a pathogen in humans. Interestingly, despite slower doubling time in older mother cells, increased virulence was observed in older mother cells from *C. neoformans*, *C. glabrata,* and *C. auris* [17,18,29]. When older cells from these three pathogens were injected in *Galleria mellonella* (waxworm) larvae, the survival rates of the larvae were significantly lower than the survival rates of the larvae infected with the young daughter cells [17,18,29]. This is consistent with the observation that older cells from these pathogenic fungi were phagocytosed at a significantly lower rate by macrophages when compared with their respective daughter cells. Macrophage- and neutrophil-mediated killings were also lower in the mother cells isolated from these pathogens [17,18,29].

One caveat is that animal experiments investigating if aging is relevant for virulence are challenging because any inoculum of old cells will multiply in vivo and the infecting cells will be predominantly young within a few duplications. However, several investigations in *C. neoformans* indicate that older cells accumulate during chronic infection. Both in an intrathecal infection model in rats as well as during chronic human infection, cells of advanced generational age could be recovered from the spinal fluid after weeks of chronic central nervous system infection. Again, this is consistent with gene expression analysis of older *C. neoformans* cells that indicated increased expression of three important established virulence factors. These factors include antiphagocytic protein (App1p) and laccase proteins (Lac1p and Lac2p) [29]. Murine experiments with *C. glabrata* have also suggested that older *C. glabrata* cells accumulate during chronic infection as evidenced by a higher bud scar count in yeast cells isolated from infected mice. Importantly, depletion of neutrophils appeared to remove the selection pressure, and as a result, the *C. glabrata* cell recovered from the infected, neutrophil-depleted mice exhibited a bud scar count suggesting that accumulation of older cells was indeed dependent on selection by resistance to neutrophil attack. This data were further supported by bud scar counts of *C. glabrata* derived from humans with Candiduria. In these patients who were chronically colonized/infected with *C. glabrata*, the median bud scar count was also higher than expected in a nonselected exponentially growing cell population [18].

## 4. Future Directions

Several studies have now demonstrated that replicative aging in several major fungal pathogens leads to important changes that affect the fungus resistance to phagocytic clearance and antifungal treatment. Therefore, the natural process of aging functions can be viewed as a process of adaptation that contributes to the pathogen’s phenotypic and genotypic variation, and subsequent resilience and virulence. The phenotypic changes of aging are not genetically inherited, and old cells only emerge because of selection pressures that promote the persistence of old cells and favor the killing of young fungal cells. Thus, for the pathogen, this form of adaptation is advantageous as it avoids the risk of random permanent mutations and instead assures that all adaptive changes are easily reversed in the daughter cells that are borne from asymmetric budding [28,52].

These findings highlight the importance of further investigating phenotypic variations that are associated with aging and convey resistance. This research can potentially identify valuable targets for drug development. Furthermore, as younger cells are more sensitive to antifungal medications and phagocytic attack, antiaging compounds may be useful drugs that can be used as combination therapy. This would pave the way for innovative and targeted antifungal therapies that could greatly add to the thinning drug pipeline. Adapting the >10-year-old MEP system from *S. cerevisiae* to pathogenic fungi by genetically modifying model strains to enable estradiol-inducible mitotic arrest of only daughter cells would permit cheap isolation of larger numbers of older cells compared with what is currently available. Ultimately, such a system could provide further novel molecular insights into why drug resistance is associated with advanced generational age.

## Figures and Tables

**Figure 1 jof-07-00006-f001:**
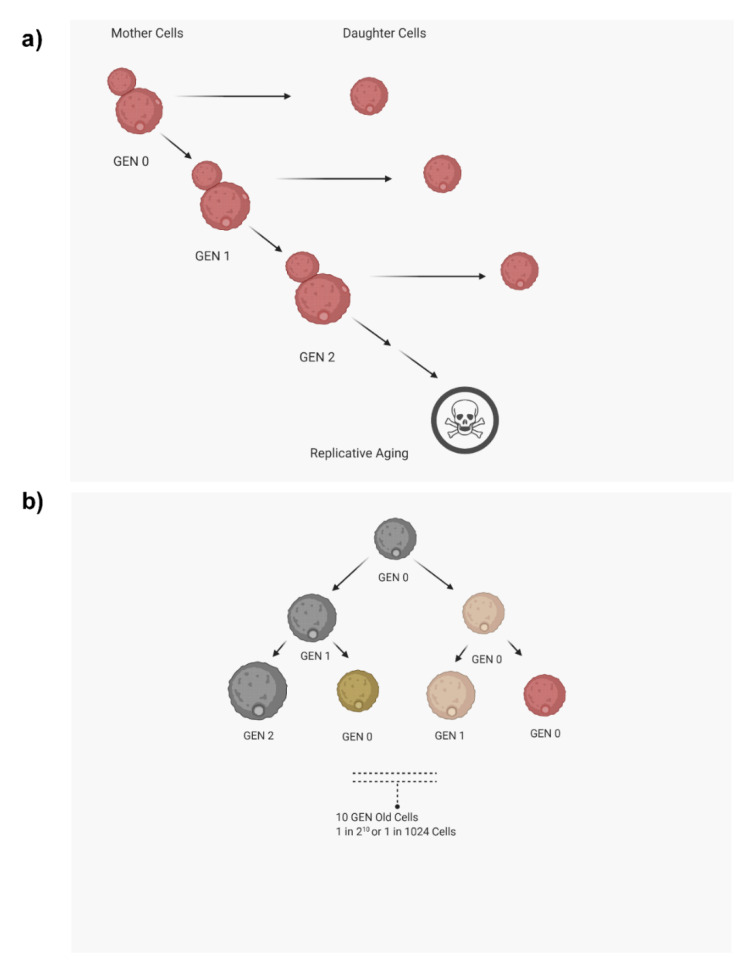
Replicative aging in yeast. (**a**) Mother cells undergo asymmetric cell division to produce daughter cells. A mother cell becomes one generation older each time it produces a bud. Each mother cell can produce a finite number of buds before it senesces. (**b**) In an exponentially growing culture, the number of older mother cells decreases with increasing age. The figure was drawn with the help of biorender.com.

**Figure 2 jof-07-00006-f002:**
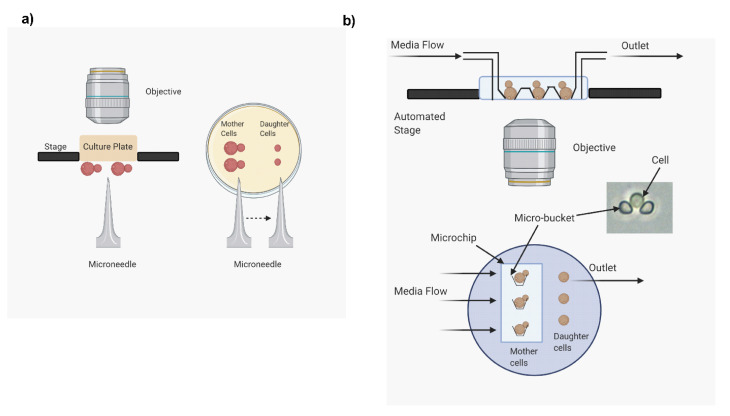
Methods for analyzing replicative life span (RLS). Two methods associated with analyzing RLS in yeast are described here. (**a**) Microdissection method: requires more manual interventions; (**b**) microfluidics method: automated and high throughput. The images were prepared with the use of biorender.com.

**Figure 3 jof-07-00006-f003:**
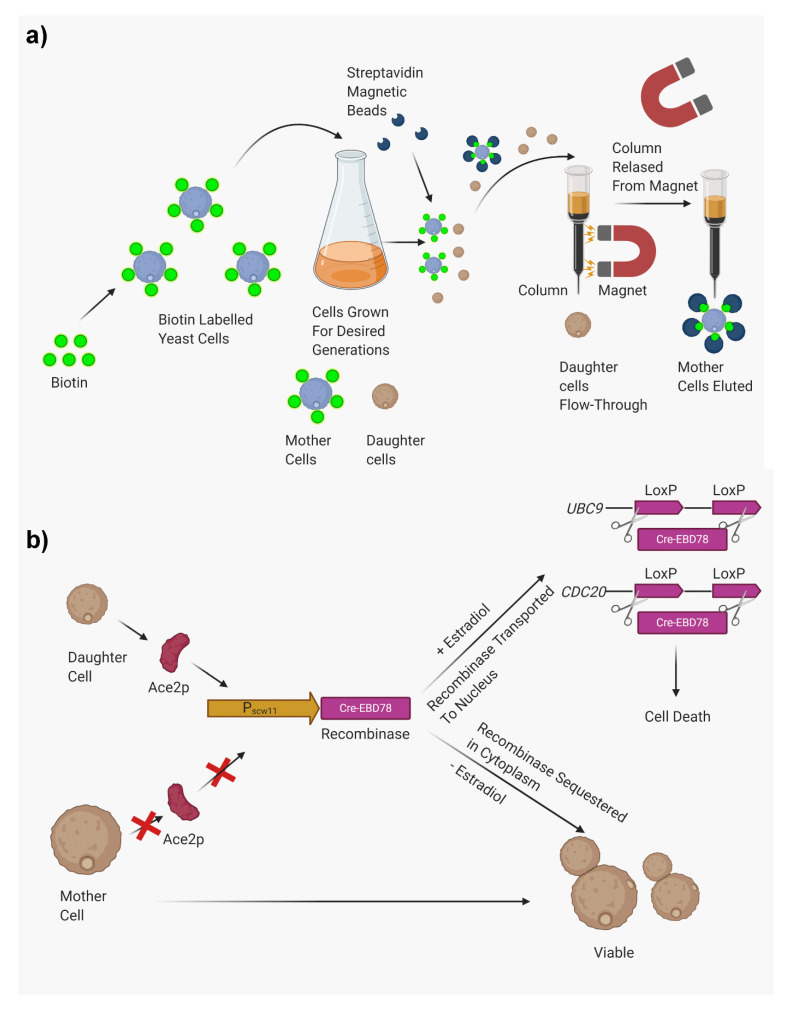
Methods for isolating generationally old mother cells. (**a**) Separation of generationally old mother cells by biotinylation and magnetic sorting. (**b**) Separation of generationally old mother cells by mother enrichment program (MEP). Figures were drawn with the help of biorender.com.

**Table 1 jof-07-00006-t001:** Methods to analyze replicative life span (RLS) used in different pathogenic yeasts.

Organism	Centrifugation	Microdissection	Biotin Label/Magnetic/FACS	High Throughput	Mother Enrichment
*Saccharomyces cerevisiae*	YES [19]	YES [19]	YES [19]	YES [12]	YES [23,28]
*Cryptococcus neoformans*	YES [21]	YES [6]	YES [21]	YES [29]	NO
*Candida albicans*	YES [20]	NO	NO	NO	NO
*Candida glabrata*	YES [18]	YES [18]	YES [18]	NO	NO
*Candida auris*	NO	YES [17]	YES [17]	NO	NO

YES signifies that these techniques were used for the respective organisms listed in the table; NO signifies that these techniques were not used for the respective organisms listed in the table.

**Table 2 jof-07-00006-t002:** Summary of phenotypes associated with increasing replicative age.

	*S. cerevisiae*	*C. neoformans*	*C. albicans*	*C. glabrata*	*C. auris*
Cell size	Increases	Increases	Increases	Increases	Increases
Cell wall	Weakens	Thickens	Thickens	Thickens	Thickens
Capsule	N/A	Increases	N/A	N/A	N/A
Cell shape	Oval	Round	Filamentous	Pseudofilamentous	Round
Cytosolic components	Increased glycogen, oxidized proteins	Not tested	Increased glycogen, oxidized proteins	Not tested	Not tested
Melanin	N/A	Increases	Not tested	Not tested	Not tested
Arrest Cell phenotype	Unbudded	Unbudded, Budded	Hyphal	Unbudded, budded, Pseudohyphal	Unbudded
Antiphagocytic ability	Not tested	Increases	Increases	Increases	Increases
Hydrogen peroxide tolerance	Not tested	Increases	Increases	Increases	Increases
Antifungal resistance	Not tested	Increases	Increases	Increases	Increases

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
