# Peer review of "Replicative Aging in Pathogenic Fungi"

_jof, 2020, doi:10.3390/jof7010006_

Round 1
Reviewer 1 Report
The most important concerm is about novelty of manuscript and being up to date. Only 3 (three)! references comes from last two years and all are autocitations. Authors used word 'recently' for paper from 2006! There are full parts in manuscript which cite the same 2-3 papers. In the case of review I would expect more recent references. In addition, partially subcject was described in authors previous papers.
Limitations of techniques listed in paper from 1998 are not surely still existing as usually mathdes are improved. If no, it should be stated.
Generally, manuscript is well written, however chapter 2.4.1 is not clear -I was not sure when the first techique decription stops and when the second starts.
I do not see the reason for citing Fig. 3 before Fig 2. in chapter 2...
Table 1-I think that references should be incorporated
Author Response
Reviewer 1:
The most important concern is about novelty of manuscript and being up to date. Only 3 (three)! references come from last two years and all are auto citations. Authors used word 'recently' for paper from 2006! There are full parts in manuscript which cite the same 2-3 papers. In the case of review I would expect more recent references. In addition, partially subject was described in authors previous papers.
Response: Thank you for the comment.
Action taken: We agree with the statement. The present studies on RA in pathogenic yeasts uses well established techniques of S. cerevisiae aging with some modifications. Most of the techniques in S. cerevisiae were established over 20 years ago. and hence, the original references are old. We have updated the references and the manuscript now includes 14 new references, which are more recent citations from the S. cerevisiae literature [1-14]. We have included a sentence acknowledging that the references are older in Line 66, which now reads as follows:
“Methods to investigate aging were first pioneered in S. cerevisiae beginning 1959 [8]. Most of these techniques are over 20 years old [8]”.
Response: please note that the data on replicative aging (RA) in pathogenic yeasts was done by Fries Laboratory except for Ref[15]. Most studies on RA aging were done in non-pathogenic Saccharomyces cerevisiae, between 1959-2020. RA results in different phenotypic variations between mother and daughter cells isolated from pathogenic and non-pathogenic yeasts. Thus, it is novel and important to study RA in pathogenic yeasts. We have added a few sentences in the introduction that states as follows:
“There is now a set of published data that indicates that older mother cells of several pathogenic fungi including Candida glabrata, Candida auris, Candida albicans, and Cryptococcus neoformans exhibit enhanced resilience to antifungals and oxidative stress, which could potentially give them an advantage in the host environment. Besides enhanced antifungal resilience, generationally aged cells from pathogenic yeasts are tolerant to phagocytic killings by macrophages. However, these phenotypic variations during RA in pathogenic yeasts are not observed in generationally aging S. cerevisiae cells [16]. Thus, it is important to study RA in pathogenic yeasts.”
Limitations of techniques listed in paper from 1998 are not surely still existing as usually methods are improved. If no, it should be stated.
Response: Thank you for the comment. The limitations listed in Module 2.4 still exists and hence this method is not commonly used anymore in studying RA in yeasts. We have added the following statement:
“The limitations of both type of centrifugation techniques still exists. Hence, presently the other techniques are preferred in studying RA in pathogenic yeasts.”
Generally, manuscript is well written, however chapter 2.4.1 is not clear -I was not sure when the first technique description stops and when the second starts.
Response: Thank you very much for the wonderful comment. We have added a sentence for better clarity. The sentence reads as follows:
“Since the above technique is labor intensive, another centrifugation method of studying RA was designed that is described below:”
I do not see the reason for citing Fig. 3 before Fig 2. in chapter 2...
Response: Thank you for the comment. We have now rearranged the figure names so that Figure 3 is cited after Figure 2 in Chapter 2.
Table 1-I think that references should be incorporated
Ans: Thank you very much for the wonderful comment. We have now added the references to the table.
Table 1. Methods to Analyze Replicative Life Span (RLS) used in different pathogenic yeasts.
|
Organism |
Centrifugation |
Micro-dissection |
Biotin label/Magnetic/FACS |
High throughput |
Mother Enrichment |
|
S. cerevisiae |
YES [17] |
YES [17] |
YES [17] |
YES [18] |
YES [19,20] |
|
C. neoformans |
YES [21] |
YES [16] |
YES [21] |
YES [22] |
NO |
|
C. albicans |
YES [15] |
NO |
NO |
NO |
NO |
|
C. glabrata |
YES [23] |
YES [23] |
YES [23] |
NO |
NO |
|
C. auris |
NO |
YES [24] |
YES [24] |
NO |
NO |
YES signifies that these techniques were used for the respective organisms listed in the table.
NO signifies that these techniques were not used for the respective organisms listed in the table.
References:
- Sarnoski, E.A.; Liu, P.; Acar, M. A High-Throughput Screen for Yeast Replicative Lifespan Identifies Lifespan-Extending Compounds. Cell reports 2017, 21, 2639-2646, doi:10.1016/j.celrep.2017.11.002.
- Chen, K.L.; Crane, M.M.; Kaeberlein, M. Microfluidic technologies for yeast replicative lifespan studies. Mechanisms of ageing and development 2017, 161, 262-269, doi:10.1016/j.mad.2016.03.009.
- Kainz, K.; Bauer, M.A.; Madeo, F.; Carmona-Gutierrez, D. Fungal infections in humans: the silent crisis. Microb Cell 2020, 7, 143-145, doi:10.15698/mic2020.06.718.
- Hao, R.O.L.a.M.J.a.Y.L.a.L.P.a.L.S.T.a.J.H.a.N. Advances in quantitative biology methods for studying replicative aging in Saccharomyces cerevisiae. Translational Medicine of Aging
2020, 4, 151-160, doi:https://doi.org/10.1016/j.tma.2019.09.002.
- Chen, K.; Shen, W.; Zhang, Z.; Xiong, F.; Ouyang, Q.; Luo, C. Age-dependent decline in stress response capacity revealed by proteins dynamics analysis. Sci Rep 2020, 10, 15211, doi:10.1038/s41598-020-72167-4.
- Azbarova, A.V.; Galkina, K.V.; Sorokin, M.I.; Severin, F.F.; Knorre, D.A. The contribution of Saccharomyces cerevisiae replicative age to the variations in the levels of Trx2p, Pdr5p, Can1p and Idh isoforms. Sci Rep 2017, 7, 13220, doi:10.1038/s41598-017-13576-w.
- Jo, M.C.; Liu, W.; Gu, L.; Dang, W.; Qin, L. High-throughput analysis of yeast replicative aging using a microfluidic system. Proc Natl Acad Sci U S A 2015, 112, 9364-9369, doi:10.1073/pnas.1510328112.
- Coody, T.K.; Hughes, A.L. Advancing the aging biology toolkit. Elife 2018, 7, doi:10.7554/eLife.42976.
- He, C.; Zhou, C.; Kennedy, B.K. The yeast replicative aging model. Biochimica et biophysica acta. Molecular basis of disease 2018, 1864, 2690-2696, doi:10.1016/j.bbadis.2018.02.023.
- Liu, P.; Acar, M. The generational scalability of single-cell replicative aging. Science advances 2018, 4, eaao4666, doi:10.1126/sciadv.aao4666.
- Ghavidel, A.; Baxi, K.; Prusinkiewicz, M.; Swan, C.; Belak, Z.R.; Eskiw, C.H.; Carvalho, C.E.; Harkness, T.A. Rapid Nuclear Exclusion of Hcm1 in Aging Saccharomyces cerevisiae Leads to Vacuolar Alkalization and Replicative Senescence. G3 2018, 8, 1579-1592, doi:10.1534/g3.118.200161.
- Pal, S.; Postnikoff, S.D.; Chavez, M.; Tyler, J.K. Impaired cohesion and homologous recombination during replicative aging in budding yeast. Science advances 2018, 4, eaaq0236, doi:10.1126/sciadv.aaq0236.
- Leupold, S.; Hubmann, G.; Litsios, A.; Meinema, A.C.; Takhaveev, V.; Papagiannakis, A.; Niebel, B.; Janssens, G.; Siegel, D.; Heinemann, M. Saccharomyces cerevisiae goes through distinct metabolic phases during its replicative lifespan. Elife 2019, 8, doi:10.7554/eLife.41046.
- Spivey, E.C.; Jones, S.K., Jr.; Rybarski, J.R.; Saifuddin, F.A.; Finkelstein, I.J. An aging-independent replicative lifespan in a symmetrically dividing eukaryote. Elife 2017, 6, doi:10.7554/eLife.20340.
- Fu, X.H.; Meng, F.L.; Hu, Y.; Zhou, J.Q. Candida albicans, a distinctive fungal model for cellular aging study. Aging Cell 2008, 7, 746-757, doi:10.1111/j.1474-9726.2008.00424.x.
- Bouklas, T.; Pechuan, X.; Goldman, D.L.; Edelman, B.; Bergman, A.; Fries, B.C. Old Cryptococcus neoformans cells contribute to virulence in chronic cryptococcosis. mBio 2013, 4, doi:10.1128/mBio.00455-13.
- Sinclair, D.; Mills, K.; Guarente, L. Aging in Saccharomyces cerevisiae. Annu Rev Microbiol 1998, 52, 533-560, doi:10.1146/annurev.micro.52.1.533.
- Crane, M.M.; Clark, I.B.; Bakker, E.; Smith, S.; Swain, P.S. A microfluidic system for studying ageing and dynamic single-cell responses in budding yeast. PloS one 2014, 9, e100042, doi:10.1371/journal.pone.0100042.
- Henderson, K.A.; Hughes, A.L.; Gottschling, D.E. Mother-daughter asymmetry of pH underlies aging and rejuvenation in yeast. Elife 2014, 3, e03504, doi:10.7554/eLife.03504.
- Lindstrom, D.L.; Gottschling, D.E. The mother enrichment program: a genetic system for facile replicative life span analysis in Saccharomyces cerevisiae. Genetics 2009, 183, 413-422, 411SI-413SI, doi:10.1534/genetics.109.106229.
- Jain, N.; Cook, E.; Xess, I.; Hasan, F.; Fries, D.; Fries, B.C. Isolation and characterization of senescent Cryptococcus neoformans and implications for phenotypic switching and pathogenesis in chronic cryptococcosis. Eukaryot Cell 2009, 8, 858-866, doi:10.1128/EC.00017-09.
- Orner, E.P.; Bhattacharya, S.; Kalenja, K.; Hayden, D.; Del Poeta, M.; Fries, B.C. Cell Wall-Associated Virulence Factors Contribute to Increased Resilience of Old Cryptococcus neoformans Cells. Front Microbiol 2019, 10, 2513, doi:10.3389/fmicb.2019.02513.
- Bouklas, T.; Alonso-Crisostomo, L.; Szekely, T., Jr.; Diago-Navarro, E.; Orner, E.P.; Smith, K.; Munshi, M.A.; Del Poeta, M.; Balazsi, G.; Fries, B.C. Generational distribution of a Candida glabrata population: Resilient old cells prevail, while younger cells dominate in the vulnerable host. PLoS pathogens 2017, 13, e1006355, doi:10.1371/journal.ppat.1006355.
- Bhattacharya, S.; Holowka, T.; Orner, E.P.; Fries, B.C. Gene Duplication Associated with Increased Fluconazole Tolerance in Candida auris cells of Advanced Generational Age. Sci Rep 2019, 9, 5052, doi:10.1038/s41598-019-41513-6.

Reviewer 2 Report
In the review entitled "Replicative aging in pathogenic fungi" the authors describe several methods that are used in RA studying, and also its influence on fungi virulence and antifungal resistance.
The paper is very well structured and easy to follow. It also points out the essential of each method used for RA determination, their advantages and draw-backs as well.
In my opinion, the paper can be accepted for publication in JoF, after a minor revision:
- page 1-line 19: replace "progressively age" with "progressively ages";
- page 1 - line 20: replace "which result" with "which results";
- page 1-line 31: replace "efficacious anti-fungal" with "effective antifungal";
- page 4-line 111: replace "unnecessary stresses" with "unnecessary stress";
- page 8-line 238: replace "other stresses" with "other type(s) of stress";
- replace "in vivo" with "in vivo";
- page8-lines 258-259: replace"different stresses" with "different types of stress";
- page 9-line 274: replace " helps maintain" with "helps maintaining";
- page 9-line 277: replace "production has been" with "production have been";
- page 9 - line 290: replace "Candida and" with "Candida and";
- more than 50% of the references cited are more than 10 years old; bibliography should be more up to date!
Author Response
In the review entitled "Replicative aging in pathogenic fungi" the authors describe several methods that are used in RA studying, and also its influence on fungi virulence and antifungal resistance.
The paper is very well structured and easy to follow. It also points out the essential of each method used for RA determination, their advantages and draw-backs as well.
In my opinion, the paper can be accepted for publication in JoF, after a minor revision:
Response: Thank you for the review and suggestions
- page 1-line 19: replace "progressively age" with "progressively ages";
Response: Thank you we revised accordingly.
- page 1 - line 20: replace "which result" with "which results";
Response: Thank you we revised accordingly.
- page 1-line 31: replace "efficacious anti-fungal" with "effective antifungal";
Response: Thank you we revised accordingly.
- page 4-line 111: replace "unnecessary stresses" with "unnecessary stress";
Response: Thank you we revised accordingly.
- page 8-line 238: replace "other stresses" with "other type(s) of stress";
Response: Thank you we revised accordingly.
- replace "in vivo" with "in vivo";
Response: Thank you we revised accordingly.
- page8-lines 258-259: replace"different stresses" with "different types of stress";
Response: Thank you we revised accordingly.
- page 9-line 274: replace " helps maintain" with "helps maintaining";
Response: Thank you we revised accordingly.
- page 9-line 277: replace "production has been" with "production have been";
Response: Thank you we revised accordingly.
- page 9 - line 290: replace "Candida and" with "Candida and";
Response: Thank you we revised accordingly.
- more than 50% of the references cited are more than 10 years old; bibliography should be more up to date!
Response: The present studies on RA in pathogenic yeasts uses well established techniques of S. cerevisiae aging with some modifications. Most of the techniques in S. cerevisiae were established over 20 years ago. and hence, the original references are old. We have updated the references and the manuscript now includes 14 new references, which are more recent citations from the S. cerevisiae literature [1-14]. We have included a sentence acknowledging that the references are older in line 66), which now reads as follows:
“Methods to investigate aging were first pioneered in S. cerevisiae beginning 1959 [8]. Most of these techniques are over 20 years old [8]”.
Response: see response to other reviewer
New References
- Sarnoski, E.A.; Liu, P.; Acar, M. A High-Throughput Screen for Yeast Replicative Lifespan Identifies Lifespan-Extending Compounds. Cell reports 2017, 21, 2639-2646, doi:10.1016/j.celrep.2017.11.002.
- Chen, K.L.; Crane, M.M.; Kaeberlein, M. Microfluidic technologies for yeast replicative lifespan studies. Mechanisms of ageing and development 2017, 161, 262-269, doi:10.1016/j.mad.2016.03.009.
- Kainz, K.; Bauer, M.A.; Madeo, F.; Carmona-Gutierrez, D. Fungal infections in humans: the silent crisis. Microb Cell 2020, 7, 143-145, doi:10.15698/mic2020.06.718.
- Hao, R.O.L.a.M.J.a.Y.L.a.L.P.a.L.S.T.a.J.H.a.N. Advances in quantitative biology methods for studying replicative aging in Saccharomyces cerevisiae. Translational Medicine of Aging
2020, 4, 151-160, doi:https://doi.org/10.1016/j.tma.2019.09.002.
- Chen, K.; Shen, W.; Zhang, Z.; Xiong, F.; Ouyang, Q.; Luo, C. Age-dependent decline in stress response capacity revealed by proteins dynamics analysis. Sci Rep 2020, 10, 15211, doi:10.1038/s41598-020-72167-4.
- Azbarova, A.V.; Galkina, K.V.; Sorokin, M.I.; Severin, F.F.; Knorre, D.A. The contribution of Saccharomyces cerevisiae replicative age to the variations in the levels of Trx2p, Pdr5p, Can1p and Idh isoforms. Sci Rep 2017, 7, 13220, doi:10.1038/s41598-017-13576-w.
- Jo, M.C.; Liu, W.; Gu, L.; Dang, W.; Qin, L. High-throughput analysis of yeast replicative aging using a microfluidic system. Proc Natl Acad Sci U S A 2015, 112, 9364-9369, doi:10.1073/pnas.1510328112.
- Coody, T.K.; Hughes, A.L. Advancing the aging biology toolkit. Elife 2018, 7, doi:10.7554/eLife.42976.
- He, C.; Zhou, C.; Kennedy, B.K. The yeast replicative aging model. Biochimica et biophysica acta. Molecular basis of disease 2018, 1864, 2690-2696, doi:10.1016/j.bbadis.2018.02.023.
- Liu, P.; Acar, M. The generational scalability of single-cell replicative aging. Science advances 2018, 4, eaao4666, doi:10.1126/sciadv.aao4666.
- Ghavidel, A.; Baxi, K.; Prusinkiewicz, M.; Swan, C.; Belak, Z.R.; Eskiw, C.H.; Carvalho, C.E.; Harkness, T.A. Rapid Nuclear Exclusion of Hcm1 in Aging Saccharomyces cerevisiae Leads to Vacuolar Alkalization and Replicative Senescence. G3 2018, 8, 1579-1592, doi:10.1534/g3.118.200161.
- Pal, S.; Postnikoff, S.D.; Chavez, M.; Tyler, J.K. Impaired cohesion and homologous recombination during replicative aging in budding yeast. Science advances 2018, 4, eaaq0236, doi:10.1126/sciadv.aaq0236.
- Leupold, S.; Hubmann, G.; Litsios, A.; Meinema, A.C.; Takhaveev, V.; Papagiannakis, A.; Niebel, B.; Janssens, G.; Siegel, D.; Heinemann, M. Saccharomyces cerevisiae goes through distinct metabolic phases during its replicative lifespan. Elife 2019, 8, doi:10.7554/eLife.41046.
- Spivey, E.C.; Jones, S.K., Jr.; Rybarski, J.R.; Saifuddin, F.A.; Finkelstein, I.J. An aging-independent replicative lifespan in a symmetrically dividing eukaryote. Elife 2017, 6, doi:10.7554/eLife.20340.

Round 2
Reviewer 1 Report
Authors corrected manuscript according to my comments.